# Chemogenetic ON and OFF switches for RNA virus replication

E. Heilmann[1,2], J. Kimpel[1], B. Hofer[1], A. Rössler[1], I. Blaas[1], L. Egerer[1,3], T. Nolden[1,3], C. Urbiola[1,2,3], H. G. Kräusslich[4,5], G. Wollmann [1,2,6 ✉] & D. von Laer[1,6 ✉]

Therapeutic application of RNA viruses as oncolytic agents or gene vectors requires a tight control of virus activity if toxicity is a concern. Here we present a regulator switch for RNA viruses using a conditional protease approach, in which the function of at least one viral protein essential for transcription and replication is linked to autocatalytical, exogenous human immunodeficiency virus (HIV) protease activity. Virus activity can be en- or disabled by various HIV protease inhibitors. Incorporating the HIV protease dimer in the genome of vesicular stomatitis virus (VSV) into the open reading frame of either the P- or L-protein resulted in an ON switch. Here, virus activity depends on co-application of protease inhibitor in a dose-dependent manner. Conversely, an N-terminal VSV polymerase tag with the HIV protease dimer constitutes an OFF switch, as application of protease inhibitor stops virus activity. This technology may also be applicable to other potentially therapeutic RNA viruses.

[1] Institute of Virology, Medical University of Innsbruck, Innsbruck, Austria. [2] Christian Doppler Laboratory for Viral Immunotherapy of Cancer, Medical University of Innsbruck, Innsbruck, Austria. [3] ViraTherapeutics GmbH, Innsbruck, Austria. [4] Department of Infectious Diseases, Virology, University Hospital Heidelberg, Heidelberg, Germany. [5] German Center for Infectious Disease Research, partner site Heidelberg, Heidelberg, Germany. [6] These authors jointly supervised this work: G. Wollmann, D. von Laer. ✉email: guido.wollmann@i-med.ac.at; dorothee.von-Laer@i-med.ac.at

Viruses have shown great potential as gene therapy vectors[1], as oncolytic viruses in cancer therapy[2] and as vaccines in humans and animals[3,4]. If replication of viruses or viral vectors could be externally regulated on demand, their individual and environmental safety would increase dramatically. While control of therapeutic DNA viruses has long been established using for example tetracycline-controlled transcriptional activation[5,6], effective external control of RNA viruses remains a challenge. In recent years, a small number of systems have been explored to provide some level of RNA regulation. Of those, RNA-aptazymes fused to a viral gene were shown to regulate virus replication over a range of <100-fold[7] and viral transgene expression up to 30-fold[8]. In addition, OFF-switch control of measles virus RNA replication was shown via small molecule-assisted shutoff (SMASh)–tags fused C-terminally to the viral P protein[9]. An externally tunable ON switch for RNA viruses has so far only been shown with photo responsive elements[10] which would be impractical for most clinical applications. None of these switches could completely block virus replication.

Vesicular stomatitis virus (VSV), a rapidly replicating negative strand RNA virus, has been widely studied as a vaccine vector[11], oncolytic virus[12], and tracing tool[13]. Although numerous VSV variants are currently under clinical development[14,15], concerns regarding potential neurotoxicity in immunocompromised hosts or in situations with exposure to the central nervous system have curbed a more rapid translational advance. In addition, its use in veterinary medicine is limited by the risk of spread among live-stock. The VSV RNA genome contains five viral genes coding for nucleoprotein (N), phosphoprotein (P), matrix protein (M), glycoprotein (G), and polymerase or large-protein (L)[16]. The genes are separated by intergenic regions that enable the transcription of several viral mRNAs from one template RNA[17].

The human immunodeficiency virus HIV protease (PR) is comprised of 99 amino acids, is active as a homodimer and functions as an aspartyl protease[18]. In HIV, it autocatalytically cleaves the polyprotein that is translated from the positive strand genome into functional proteins. HIV PR activity can be regulated by several clinically approved protease inhibitors (PIs) that are widely used in HIV therapy.

Here we present a control mechanism for RNA viruses by fusing the autocatalytically active HIV protease dimer and its corresponding cleavage sites into or adjacent to essential viral proteins of VSV. Adding PI compounds either enabled or blocked virus replication, depending on the fusion design. The intramolecular insertion of the protease dimer in either the P or L protein of VSV constituted a functional ON-switch. In contrast, N-terminal fusion to L resulted in an OFF-switch when PI was applied. We performed in vitro and in vivo studies to confirm the dose-dependent and robust regulation of both, viral replication as well as viral transgene expression. The conditional proteolytic switch system allowed the control of RNA-virus replication from complete inhibition to nearly wild-type replication levels.

## Results

**Generation and in vitro functionality of ON-switch viruses.** We first incorporated the PR dimer (prot) into the viral P protein[19], flanked by corresponding cleavage sites[20] and flexible linkers[21] (Fig. 1a and Supplementary Figs. 1a and 2a). To initially test for control of P-protein function in a mini-genome assay, the PR dimer was cloned into a P-expression plasmid at amino acid position 196 of P, which was previously reported to tolerate intramolecular insertions[22] (Pprot). Cells transfected with this plasmid and co-infected with a P-protein-deficient VSV variant encoding DsRed[23] showed red fluorescence in the presence, but not absence of the PI amprenavir (APV) (Supplementary Fig. 1b).

This result indicates that the P fusion protein was able to functionally engage in the N-P-L replication complex only when PR was inactivated (Supplementary Fig. 1C). Next, the Pprot ON-switch construct was cloned into the VSV genome, generating VSV-Pprot-GFP (Fig. 1a). GFP expression, cytopathic effects, and viral replication were only observed in the presence of APV with dose-dependent regulation of VSV replication in the range from 300 nM to 100 µM APV (Fig. 1g, h). Conversely, in the absence of APV, P would be cleaved and the replication complex disassembled (Fig. 1b). Other clinically used PIs including saquinavir (SQV, 10 µM), lopinavir (10 µM), and indinavir (10 µM) also regulated VSV-Pprot-GFP (Supplementary Fig. 1e). Western blotting against HIV protease confirmed the proteolytic cleavage in the absence of PI with a band marking the HIV protease dimer at ~22 kDa[19], compared to a band at 54 kDa corresponding to the Pprot fusion protein in the presence of PI (Fig. 1c).To address genetic stability, we performed in vitro serial virus passages in optimal (10 µM) and suboptimal (1 µM) amprenavir conditions. After each passage protease inhibitor dependency was assessed by GFP expression in the presence or absence of PI (Supplementary Fig. 3a). After 20 passages (P20), no amprenavir escape virus variants could be observed and functional PI dependency was confirmed by GFP fluorescence and viral plaque assay (Supplementary Fig. 3b). A protease-insert-spanning PCR was used to confirm the presence of a single band matching the size for an intact PR dimer insert (Supplementary Fig. 3c). Subsequent sequencing of the P-prot insert (not the whole viral genome) and alignment comparison with the parental plasmid construct revealed one mutation (protease 2: nucleotide G23A; amino acid R8K) in the construct with no effect on the proteolytic control of virus activity (Supplementary Fig. 3d). To test if the regulatable proteolytic switch also works on another element of the viral replication complex, we inserted the PR dimer into the VSV L-protein at amino acid position 1620 previously identified by our group[24]. The resulting recombinant virus VSV-Lprot-GFP (Fig. 1d and Supplementary Fig. 1d) showed similar characteristics as VSV-Pprot-GFP and responded to APV in vitro in a comparable dose-dependent manner (Fig. 1i). Consequently, western blot analysis revealed an intact large L protein HIV protease dimer fusion protein of ~266 kDa in the presence of PI and a separated HIV protease dimer band of ~22 kDa in the absence of PI (Fig. 1e, f). Sequencing of VSV-Lprot-GFP revealed secondary mutations in L (Supplementary Fig. 4), that had emerged during virus rescue, analogous to findings in our previous study[24] suggesting that these mutations could be essential for the efficient replication of L-insertion VSV. We next compared the single step growth curve kinetics of the two regulatable virus variants with the parental VSV-GFP. In the presence of 10 µM APV VSV-Lprot-GFP showed no significant attenuation in replication kinetic while VSV-Pprot-GFP showed mild attenuation in the range of 1–1.5 logs compared to parental viral titers (Fig. 1j). Analogous to the Pprot variant above, 20-fold serial passage in the presence of both optimal (10 µM) and suboptimal concentrations (1 µM) of APV did not produce any viral progeny lacking dependency on protease inhibition (Supplementary Fig. 5). However, as a proof-of-concept to further minimize the risk for potential reversion to wild-type VSV, a tandem insertion was made combining the PR dimer insertions into P and L. We had confirmed feasibility of this approach by generating a recombinant VSV with two fluorescent protein insertions, one in P and one in L[24]. Based on this observation, a recombinant VSV with PR dimer insertions in P and L (VSV-P-Lprot-GFP) was generated (Supplementary Fig. 6a, b). This virus also showed PI dependency, expressing GFP and generating small plaques only in the presence of APV (Supplementary Fig. 6d). However, the dual switch construct was significantly attenuated (>2 log reduction in

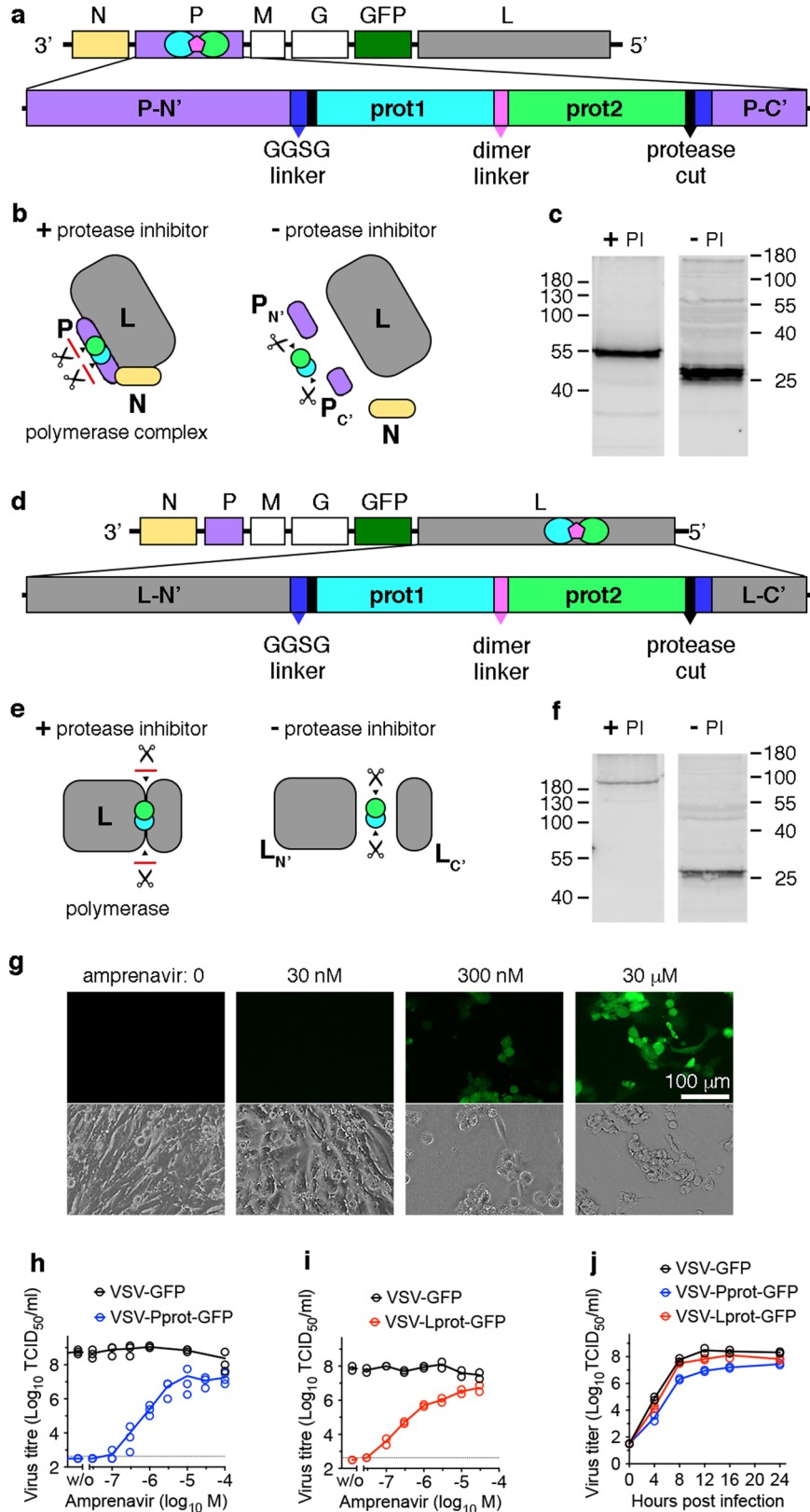

virus titers) compared to single insertion variants and was not further investigated in this study.

**ON-switch viruses are regulatable in vivo**. To validate the ON-switch viruses in vivo, we generated a luciferase-expressing variant VSV-Pprot-Luc. APV controlled bioluminescence activity of cells infected with this virus in a dose-dependent manner as well (Supplementary Fig. 7a). Intratumoral application of $2 \times 10^6$ infectious units (titrated via $TCID_{50}$) VSV-Pprot-Luc into subcutaneous U87 xenografts in nude mice resulted in an initial bioluminescence signal independent of PI application. This result

**Fig. 1 VSV-Pprot and VSV-Lprot require the presence of HIV protease inhibitor for virus activity. a** Scheme of the VSV genome with detailed depiction of the intramolecular insertion into the phosphoprotein (P) of the HIV protease linked-dimer construct flanked by protease cleavage sites and GGSG linker sequences. **b** Model of the intact VSV nucleoprotein-phosphoprotein-polymerase (N-P-L) replication complex in the presence of protease inhibitor. Absence of protease inhibitor results in autoproteolytic separation of P into N- and C-terminal fragments ($P_{N'}$, $P_{C'}$) and disengagement from the replication complex. **c** Western Blot against HIV protease display the large fusion protein in presence of protease inhibitor (+PI) resolved prior on low percentage polyacrylamide gel. Protein samples without protease inhibitor (-PI) were produced on replication-supporting 293-VSV cells expressing N, P, and L and resolved on a high percentage polyacrylamide gel. **d** Scheme of the intramolecular insertion into the VSV L protein of the same protease dimer construct. **e** Model of the intact VSV L protein in the presence of protease inhibitor. Absence of protease inhibitor results in autoproteolytic separation of L into two fragments ($L_{N'}$, $L_{C'}$) and halted polymerase activity. **f** Western Blot against HIV protease with protease inhibitor (+PI) from large-pore gel and without protease inhibitor (-PI) from small-pore gel. Virus replication on BHK cells leading to GFP viral reporter gene expression (**g**) required the presence of PI amprenavir (APV) in a dose-dependent manner for VSV-Pprot-GFP (**h**) and VSV-Lprot-GFP (**i**). BHK cells were infected with VSV-GFP, VSV-Pprot-GFP, or VSV-Lprot-GFP at an MOI of 1 in the presence of increasing doses of APV. Supernatant was collected 24 h later and virus progeny titer was determined via $TCID_{50}$ assay (technical replicates; $n = 2$). **j** Replication kinetic of VSV-GFP vs. VSV-Lprot-GFP and VSV-Pprot-GFP. BHK cells were infected with VSV-GFP, VSV-Lprot-GFP, or VSV-Pprot-GFP at an MOI of 3 in the presence of 10 µM APV. Supernatants were collected at indicated time points and virus progeny titer was determined via $TCID_{50}$ assay (technical replicates; $n = 3$; dotted line indicates detection limit). Western blot experiments in panels (**c**, **f**) were performed three times. Viral growth curve studies in panels (**h**, **i**, **j**) were performed two times times. Source data are provided in the Source Data File.

can be explained by the presence of APV remnant from the virus preparations to block autoproteolysis during virus production and storage. Without further PI injections, the bioluminescence signal decreased significantly after 3 days (Fig. 2b) followed by loss of tumor control (Supplementary Fig. 7b). In contrast, in PI (APV + ritonavir (RTV)) -treated mice, luciferase signal plateaued for 17 days (Fig. 2a, b) and tumors were controlled in size (Supplementary Fig. 7b), confirming the in vivo functionality of the ON switch construct. Intratumoral treatment of U87 xenografts with the VSV-Lprot-GFP variant resulted in a significantly delayed tumor growth and survival benefit in the presence of PI (APV + RTV) compared to treatment without the compound mix (Fig. 2c, d), further validating the in vivo applicability of the virus switch. Importantly, virus isolated from select tumors maintained PI dependency in vitro and sequencing of the insert region of either VSV-Pprot-Luc or VSV-Lprot-GFP virus isolates after in vivo passage revealed no change in sequence compared to the start virus, indicating a robustness of the regulatable construct.

Wild-type-based VSV is known for pronounced neurotoxicity in laboratory animals[25,26]. To address whether neurotoxicity of VSV-Pprot-GFP is abrogated compared to parental VSV variants, we employed direct stereotactic injection into the mouse striatum. Intracranial instillation of wild-type-based VSV-DsRed ($2 \times 10^5$ $TCID_{50}$ in 2 µl) led to profound signs of neurotoxicity starting 2 days post injection (Fig. 2e, f). All mice had to be euthanized within 4 days (Fig. 2g). In contrast, injection of VSV-Pprot-GFP was well tolerated with no signs of neurological abnormalities both in the presence or absence of PI (APV + RTV) co-treatment for an observation period of 9 days (Fig. 2e, f). Histological fluorescence analysis of brains harvested on the day of toxicity-related euthanasia after VSV-DsRed injection revealed extended spread of the virus through the brain parenchyma including midline crossing (Fig. 2h). In contrast, GFP expression from VSV-Pprot-GFP was restricted to the immediate lining of the injection needle track without any signs of further intracranial spread regardless of whether PI compound mix was systemically applied or not. As the CNS penetrability of APV is low[27], we performed a complementary neurotoxicity study with the VSV-Lprot-GFP virus applying an alternative PI treatment regimen with indinavir (combined with RTV; this combination has been shown to have a higher CNS penetration[28]) as well as providing 10 µM APV directly in the stereotactic injection volume. Despite the initial presence of PI in the injectate and the use of a PI regimen with higher expected CNS concentration, no signs of neurotoxicity or continued weight loss were observed (Fig. 2i and Supplementary Fig. 7c), and the virally induced GFP signal was still confined to the injection site (Fig. 2j).

**Generation and in vitro functionality of OFF-switch viruses.** Following generation and validation of a PR-based ON-switch, we attempted to produce a VSV variant that can be switched OFF by PI addition. Here, viral replication should be unrestricted in the absence of PIs, and should be blocked by addition of the compound. For this OFF-switch, the PR dimer with a variant codon optimization (Supplementary Fig. 2b) was inserted into the VSV genome to create a fusion protein of GFP, the PR dimer and the viral polymerase L (Fig. 3a). This large fusion protein is rendered functionally inactive, but can be activated by proteolytic release of L in the absence of PI (schematically shown in Fig. 3b). Western blotting against HIV protease displayed a large fusion protein, matching the combined size of the VSV polymerase, GFP, and the HIV protease dimer (~290 kDa) (Fig. 3c). In addition, homogenously distributed (free) GFP fluorescence was observed in cells infected with this variant in the absence of PI, while only weak and spotty GFP fluorescence reminiscent of VSV replication foci[29] was observed in the presence of PI, probably due to residual initial activity of free L protein included in the incoming virus at high multiplicity of infection (Fig. 3d). Replication of this recombinant virus was inversely dependent on PI dose, and could be blocked by PIs SQV (Fig. 3e) and APV (Supplementary Fig. 8a). Single step replication growth kinetics revealed no attenuation of the Prot-OFF construct compared to the parental VSV-GFP (Fig. 3f). Interestingly, various Prot-OFF constructs (differing in linker construction) also developed secondary mutations (after virus rescue and prior to any PI contact) within the first or second protease sequence at amino acid positions 85 or 86 (Supplementary Fig. 8b). These mutations are not associated with known HIV PI resistance regions and possibly made the protease more active when fused between GFP and L. Virus without these mutations could not be rescued. Consistent with the observed genetic stability with the ON switch constructs, repetitive in vitro passage over 20 cycles did not change the responsiveness of the OFF switch (Supplementary Fig. 9).

**OFF-switch viruses are regulatable in vivo.** To test Prot-OFF in vivo, subcutaneous G62 xenografts grown in NOD-SCID mice were treated with intratumoral injections of $2 \times 10^7$ $TCID_{50}$ of parental VSV-GFP ($n = 8$) or VSV-Prot-OFF ($n = 16$) at day 0 and 7. Starting day 6, mice treated with VSV-GFP showed signs of neurotoxicity (Fig. 3g). 15 days post treatment, after the first among the mice treated with VSV-Prot-OFF developed neurological symptoms, the remaining mice were randomly divided into 2 groups. One group ($n = 7$) received sole vehicle allowing continuous virus replication, which maintained tumor control but also led to further neurotoxicity in some mice. However,

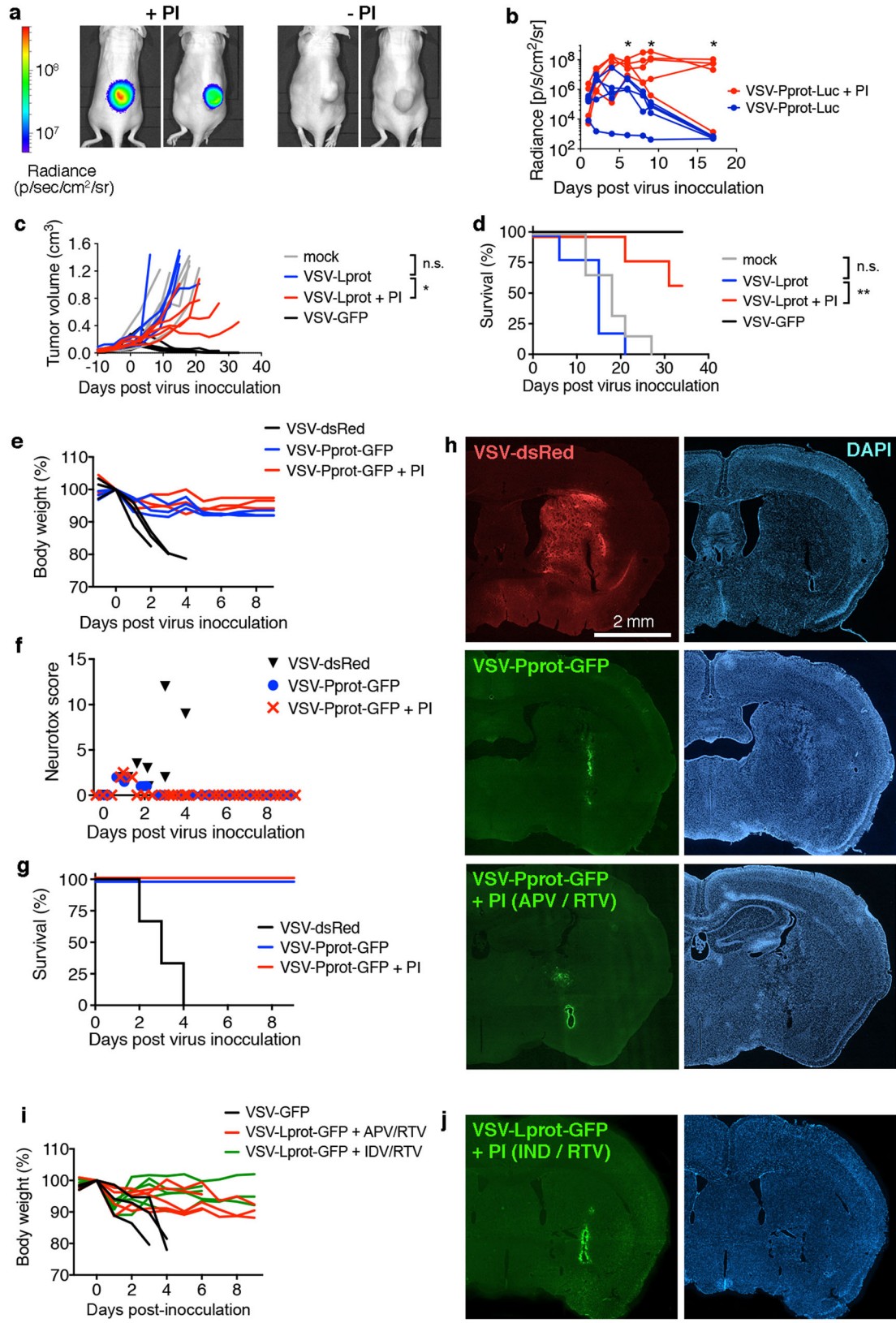

neurotoxicity was reduced compared to parental VSV-GFP – treated mice (3 vs 6 out of 8 mice). The second group ($n = 8$) was subsequently treated with a PI cocktail (SQV + RTV) 3x a day to initiate the OFF switch. No signs of neurotoxicity were observed in this group. After OFF switch activation tumor control was diminished and relapse occurred (Fig. 3g). In a parallel study,

tumors treated with a single injection of VSV-GFP or VSV-Prot-OFF with or without PI co-treatment were harvested at 3, 7, and 10 days post virus inoculation. Histological analysis of VSV-Prot-OFF infected tumors revealed intratumoral spread, although diminished relative to wild-type VSV (Fig. 3h, i), suggesting that VSV-Prot-OFF is attenuated to some extent in vivo. In contrast,

**Fig. 2 VSV-Pprot and VSV-Lprot are regulatable in vivo. a** Subcutaneous human glioma U87 xenografts were intratumorally treated with luciferase-expressing VSV-Pprot-Luc with (+PI) or without (-PI) concomitant intraperitoneal (i.p.) application of APV. Representative bioluminescence (BLI) images are shown from 8 days after virus injection. **b** BLI quantification of luciferase signal from VSV-Pprot-Luc treated tumors in mice receiving PI cocktail (APV + RTV) (red) or drug vehicle (blue) (technical replicates; $n = 5$; * denotes significantly different measurements with $p < 0.05$; unpaired two-tailed $t$ test at indicated time points). **c, d** Single intratumoral treatment of subcutaneus U87 tumors with VSV ($n = 6$) or VSV-Lprot-GFP with ($n = 5$) or without ($n = 5$) PI treatment (APV + RTV) (mock $n = 6$). Log-rank (Mantel-Cox) test was performed (**$p = 0.0052$). **e–h** 2 μl of VSV-DsRed or VSV-Pprot-GFP were stereotactically injected into the striatum of BALB/c mice. PI (APV + RTV) treatment was applied every 12 h for 9 days. VSV-Pprot-GFP was well tolerated with no significant weight loss (**e**) or signs of neurotoxicity (**f**) compared to fatal neurotoxicity of parental VSV-DsRed (**g**) (**$p = 0.0078$; Log-rank (Mantel-Cox) test). Symbols in **f** display score per mouse per time point). **h** Histological fluorescence analysis of coronal brain sections revealed extended spread of VSV-DsRed in the striatum, subcortical areas and hypothalamus (bilateral) at 3 days post inoculation (dpi). In contrast, GFP expression from VSV-Pprot-GFP (10 dpi) was restricted to the immediate lining of the injection needle track without any signs of intracranial spread, irrespective of i.p. co-treatment of PI or drug vehicle. PI treatment with high-dose IDV to increase CNS availability after VSV-Lprot-GFP injection did also not induce signs of neurotoxicity and brain parenchymal spread was restricted to the injection site (**i, j**); $n = 3$ for VSV-GFP, $n = 5$ for Lprot variants. Bioluminescence experiments in panels (**a**, **b**) and associated tumor growth and survival study (**c**, **d**) were performed once. Intracranial injection experiments (**e–h**) were performed two times. Source data are provided in the Source Data File.

PI treatment starting 3 days after virus inoculation abrogated spread of the virus, which was limited to a minor isolated region (Fig. 3j).

## Discussion

We present here a regulatory switch putting the activity of a rhabdovirus under conditional control of an exogenously applied clinically approved small molecule. In the first configuration, the so-called ON-switch, virus replication is dependent on the presence of the compound. By changing the insertion site of the switch from an INTRA- to an INTER-molecular location, we can flip the effect of the compound from an ON- to an OFF-switch.

For optimal virus activity control, both ON and OFF switches were designed to interfere with an early stage of virus propagation by modulating proteins of the viral replication/transcription machinery. Hence our approach expands and complements the effectiveness and range of control of current methods of RNA virus regulation. In comparison to the aptazyme approach to control measles virus for example, aptazymes had to be placed into both 3′- and 5′ UTRs of the viral fusion protein to achieve effective inhibition of viral spread, resulting in reduction of viral progeny by 3 logarithmic orders. Inhibition was applied at a late stage of the virus replication cycle, i.e., fusion. Apparently, even small amounts of fusion protein are sufficient to facilitate virus spread. Therefore, the reduction of three logarithmic orders was only accomplished with a multistep infection assay (low multiplicity of infection (MOI) of 0.0001) and a long observation period of 8 days. Single insertions of aptazymes in either 3′- or 5′ UTRs did not drastically reduce titers. Further development of aptazyme-controlled virus and transgene activity comprised a guanine-responsive switch in 3′ UTRs of glycoprotein-deficient VSV expressing GFP. The strongest transgene repression of 26.8-fold was facilitated by placement of the switch in both 3′ UTRs of GFP and the VSV polymerase in the presence of 500 μM guanine[8]. However, as discussed therein guanine's low solubility impedes in vivo applications. Our system, on the other hand, is based on clinically approved and extensively deployed PIs with potent control of viral replication in the low micromolar range. Such doses are comparable to dose equivalents of oral treatment regimens in human for amprenavir, saquinavir or indinavir, respectively[30,31] or precursors[32], respectively. Arguably, as the potential control of therapeutic viruses encompasses a much shorter duration compared to the chronic application of HIV PIs, the application of even higher doses of PIs might be conceivable. Of note, our in vitro dose response studies revealed that higher doses of PI are required compared to their reported EC50 concentrations for in vitro control of HIV[18]. However, the APV and SQV doses applied intraperitoneally to study controllability

in vivo studies were in line with reported human plasma levels under standard therapy[33,34]. Our presented ON and OFF switch system focuses on viral replication control and therefore transgene expression levels are linked to viral replication. Conversely, the guanine aptazyme approach enables the control of transgenes independent from the virus, which might present an advantage over our system in certain applications. Hypothetically, the combination of RNA- and protein-based regulation approaches could hold potential for future developments of regulatable viruses, e.g., to regulate virus replication and transgene expression by different small molecules or to have multiple, completely different safety mechanisms in place as fail safe. Lastly, another advantage of RNA-based regulation via aptazymes is the independence from additional xenogenic proteins within the vector, which can be immunogenic or attenuate the virus.

In theory, the ON-switch system could also present as an additional environmental safety element, although not tested in this study. As virus progeny depend on the presence of PI, potentially shed virus is not active for productive infection of a new host. This could be of importance, when therapeutic or vaccine RNA viruses can cause or mimic notifiable animal diseases, as was recently discussed in a quantitative risk assessment estimate for oncolytic treatments with Seneca Valley virus (SVV)[35]. Of note, while not being directly a serious pathogen for lifestock, wild-type VSV infection clinically mimics foot-and-mouth disease, and therefore exposure to animals should be limited[36,37].

Neurotoxicity of wild-type based VSV has been abrogated in recent years by genetic engineering for chimeric variants[38,39] or by naturally occurring relatives within the rhabdovirus family[40] with some of these variants having already entered clinical testing[41]. Therefore, to address potential toxicities of replication-competent viral therapeutics in principle and to test our regulatable ON and OFF switch constructs in vivo, we had to focus on the wild-type VSV backbone, which reproducibly elicits neurotoxicity. However, application of wild-type-based VSV in the athymic nude mouse model did not elicit neurotoxicity, consistent with intact NK cell activity in these animals and corresponding incomplete immune deficiency. We therefore employed an alternative xenograft model for testing the ability to reduce neurotoxicity after activating the OFF switch in VSV-Prot-OFF-GFP: this made use of the highly immune deficient NOD-SCID host[38]. In this model, we observed clear neurotoxicity, which could be successfully prevented when triggering the OFF switch as late as 15 days after the oncolytic treatment start. The most consistent way to trigger neurotoxicity in mice, is the direct intracranial instillation of VSV. With our conditional proteolysis ON-switch approach, another factor comes into play, however.

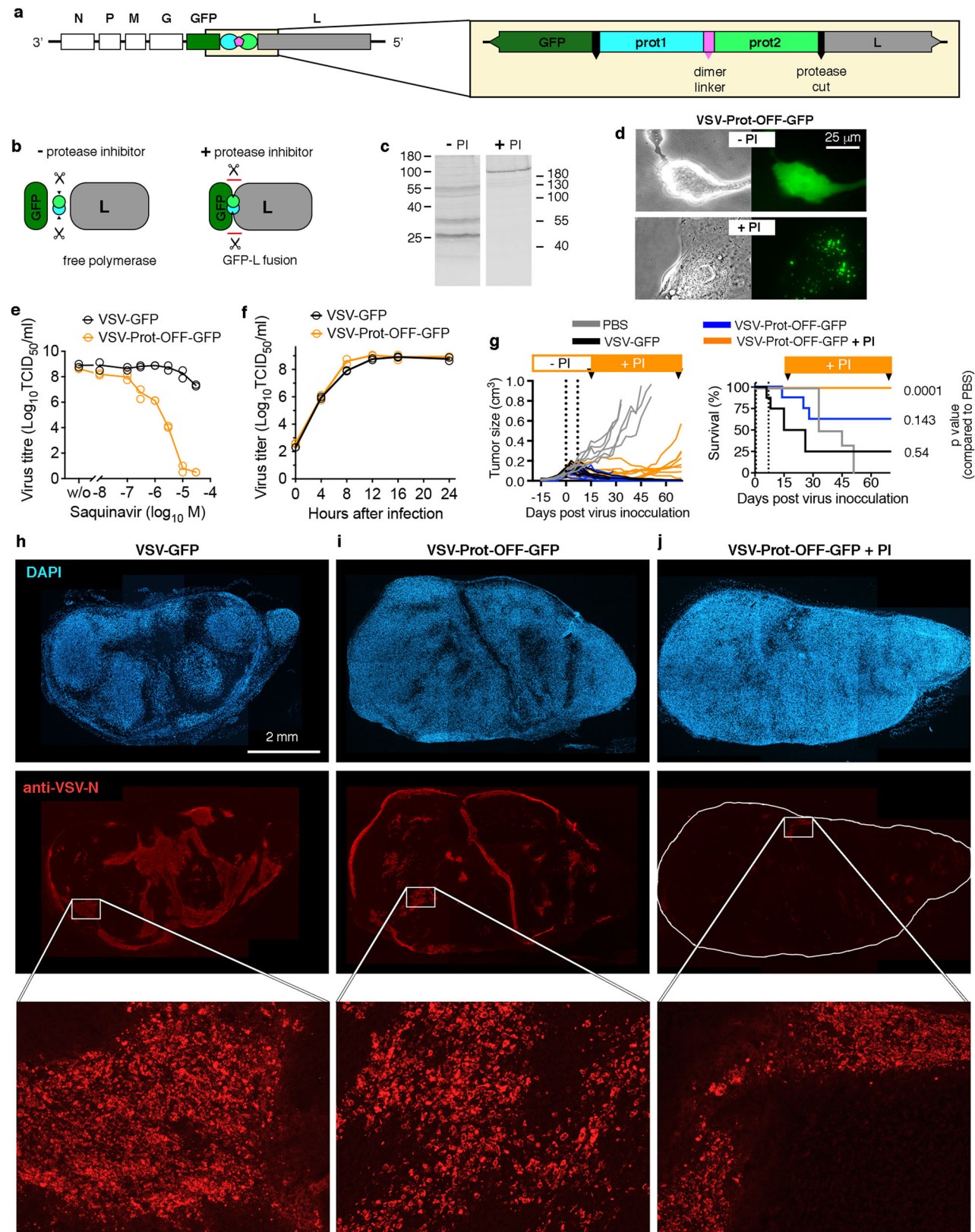

Most, if not all, HIV protease inhibitors show only limited penetration into the CNS[27], thus not allowing PI-dependent activation of viral replication in the CNS. Even applying a PI with higher CNS availability, and adding PI to the injectate, no neurotoxicity and viral spread could be observed in our experiments. We believe this actually adds an additional layer of safety to the system, as the PIs required to maintain activity of the ON switch constructs and thus mediate neurotoxicity only reach a fraction of their serum levels in the central nervous system[27]. This might be of importance for attenuated VSV variants, that are safe with peripheral application but have shown toxicity potential once entered into the brain[42].

**Fig. 3 N-terminal fusion of protease dimer construct to VSV L protein constitutes a regulatable OFF-switch. a** Scheme of the VSV genome with detailed depiction of the N-terminal fusion construct of the linked-dimer protease construct flanked by autoproteolytic cleavage sites with the L protein. **b** Scheme of the autoproteolytic detachment of the protease dimer construct and GFP from the polymerase (L) in absence of protease inhibitor and generation of fusion protein leading to GFP viral reporter gene expression in infected BHK cells in the absence of PI. **c** Western Blot against HIV protease display the released HIV protease dimer in the absence of protease inhibitor (-PI) resolved prior on high percentage polyacrylamide gel. Protein samples with protease inhibitor (+PI) were produced on replication-supporting 293-VSV cells expressing N, P, and L and resolved on a low percentage polyacrylamide gel. Treatment with PI (here saquinavir SQV, 10 μM) results in dysfunctional GFP-prot-L fusion. **d** The GFP-prot-L fusion protein can be visualized by infection of cells with high multiplicity of infection and addition of PI. Incoming virions carry functional L that produces the large fusion protein, which accumulates in typical rhabdovirus replication foci (lower panel). Without PI, GFP fluorescence is diffuse, infected cells round up and die (upper panel). **e** Dose response of VSV-GFP and VSV-Prot-OFF-GFP for protease inhibitor saquinavir (SQV). BHK cells were infected with VSV-GFP or VSV-Prot-OFF-GFP at an MOI of 1 in the presence of increasing doses of SQV. Supernatant was collected 24 hrs later and virus progeny titer was determined via $TCID_{50}$ assay ($n = 2$; technical replicates). **f** Replication kinetics of VSV-GFP vs. VSV-Prot-OFF-GFP. BHK cells were infected with VSV-GFP or VSV-Prot-OFF-GFP at an MOI of 3. Supernatants were collected at indicated time points and virus progeny titer was determined via $TCID_{50}$ assay ($n = 3$; technical replicates). **g** NOD-SCID mice bearing subcutaneous human glioma G62 xenografts were treated intratumorally twice with 7 day interval (dotted line) with either VSV-GFP ($n = 8$), VSV-Prot-OFF-GFP without ($n = 8$) or with ($n = 8$) PI treatment (SQV + RTV) 3x a day. Orange bar indicates start and duration of PI treatment. Left panel shows individual tumor growth kinetics. Right panel depicts the Kaplan–Meier survival curves (Log-rank (Mantel-Cox) test, $p$ values from comparison to PBS control). In a parallel study, PI (SQV + RTV) treatment was initiated 3 days after single virus injection and tumors treated with VSV-GFP or VSV-Prot-OFF-GFP ($n = 3$) were harvested 1 week later and analyzed for virus spread using anti-VSV immunofluorescence staining. Representative images show wide intratumoral spread of VSV-GFP (**h**), disseminated spread of VSV-Prot-OFF-GFP without SQV (**i**) and isolated reduced virus staining of VSV-Prot-OFF-GFP under SQV treatment (**j**). Western blot experiments in panel (**c**) were performed three times. High-resolution fluorescence microscopy experiments in panel **d** were repeated twice with 2 wells per condition. 10 positions of each well were monitored automatically for up to 18 h post infection. Viral growth curve studies in panels (**e**, **f**) were performed two times times. Tumor growth and survival study in panel (**g**) were performed once. Source data are provided in the Source Data File.

Using the same autoproteolytic system but in a functionally different genome location, we also developed the reversal mechanism of protease-dependent OFF regulation, which works by the replacement of an intergenic region with the HIV protease dimer. In this construct the protease must be active to separate two viral proteins, similar as it does in HIV. Adding protease inhibitor in this construct leads to non-functional fusion proteins, which inhibit viral activity. The wild-type like characteristics both in vitro and in vivo suggest that this OFF Switch is attractive for therapeutic viruses, as in oncolytic virus therapy or in gene therapy with persistent viral vectors or replicons. Here the drug would only be given in the case of viral or transgene toxicity. Thus, the presented control mechanisms might prove particularly valuable for viruses that have residual toxicity or are armed with therapeutic cargos with a small therapeutic window, such as bispecific T-cell engagers (BiTEs), IL-12, TNF-α and others. Resembling in principle the SMASH-tag[9], an advantage over this system is that the transgene (in this case GFP) is built into the switch, meaning a potential loss of the switch would necessarily entail the deletion of the potentially toxic transgene as well, i.e., would be self-disarming.

Although not the focus of the current study, we hypothesize that the presented regulatory principles can generalize to viruses other than VSV if certain conditions are met. As a central precondition to generate Prot-ON RNA viruses, target genes should be an essential part of the viral replication complex and be permissive for intramolecular insertions of transgenes. Such insertion sites have been described for several viruses, e.g., measles virus[43], rinderpest virus[44], canine distemper virus[45], Ebola virus[46], or rabies virus[10]. Insertion herein of the HIV protease, or other proteases amenable to autoproteolytic activity, should generate a regulatable Prot-ON virus similar to the presented VSV-Pprot. Conversely, the Prot-OFF principle of virus activity control should also work for other viruses when two stipulations are met. For one, replication-essential genes need to tolerate minor N- or C-terminal tags comprised of the residual amino acids of the protease recognition sites after cleavage. However, without such proteolytic cleavage these gene products would have to remain non-functional fusion constructs in the presence of large tags such as fluorescence proteins[47,48]. While numerous fluorescent protein fusion tags have been described on viral structural proteins[48,49], studies on terminal fusions to proteins involved in viral replication are limited[50]. Here, additional studies might be necessary to evaluate a virus' suitability for the OFF switch.

Taken together, the Prot-OFF and Prot-ON systems presented here allow effective control of the oncolytic virus VSV in the host by several drugs that have a long-term clinical safety record. Our chemogenetic proteolysis approach might be used for other therapeutic or vaccine viruses as well. Thus, this technology can potentially support a broad range of applications and provides the basis for future developments of safer vaccines and virotherapies.

## Methods

**Plasmids and viruses.** The coding DNA sequence for the phosphoprotein with a linked-dimer protease in position aa196 and the flanking sequences of the VSV N and M (N-Pprot-M) were synthesized by GeneArt/Thermo Fisher Scientific (Regensburg, Germany). The protease linked-dimer construct was flanked by $(GGSG)_3$ linker sequences to allow spatial separation between the P and the protease dimer (Supplementary Fig. 2a). P expression plasmid was cloned by digestion of a standard pUC19 expression vector (GenBank # AJ318514) and the insert-containing vector (GeneArt) with XbaI and Bst1107l followed by ligation with a T4 ligase (NEB, Frankfurt, Germany). In all, 293T cells were transfected with this Pprot construct using TransIT®-LT1 transfection kit (Mirus Bio LLC, Madison, WI, USA) and infected with a VSV-ΔP variant[23]. The VSV-ΔP was equipped with a red fluorescent protein as reporter gene.

Wildtype-based recombinant VSV Indiana strain and VSV-GFP were described previously[38,51]. To generate regulatable VSV variants, the N-P-M genome part of a VSV Indiana P[52] was replaced by N-Pprot-M (synthetized by GeneArt). GFP at position 5 was used as marker gene. All PCR primers are listed in detail with sequence and target region in Supplementary Table 1. N-Pprot-M was amplified by PCR (primers N-35nt-before-BstZ17I and P-35nt-after-XbaI) containing sequences overlapping 2 restriction enzyme sites (XbaI and Bst1107l) to the full-length VSV vector. The construct was then generated by Gibson assembly cloning, resulting in the ON switch VSV-Pprot-GFP. VSV-luciferase virus was generated by cloning a *photinus pyralis firefly* luciferase cassette, amplified via PCR (primer pair XhoI-luc-for/NheI-luc-rev) from pGL4.51 (Promega #E1320) via XhoI/NheI restriction sites. VSV-Pprot-luciferase was cloned analogously to the VSV-Pprot-GFP variant using the VSV-luciferase vector. To generate regulatable VSV-Lprot, the linked protease dimer construct was inserted at L protein position 1620 by four-fragment Gibson assembly as previously described[24]. In brief, the large vector fragment was provided by restriction enzyme digestion with enzymes SfoI and FseI of VSV-GFP (fragment 4). The linked protease dimer (fragment 1) was PCR amplified by primers GGSG-prot-for and -rev. L-protein sequences surrounding fragment 1 that were lost by vector digestion (fragments 2 and 3), were generated via PCR on a VSV vector. Fragments 2 and 3 received overhangs to the $(GGSG)_3$ linker at the 5′ end with

primer L-1620-GGSG-rev and 3′ end with primer L-1620-GGSG-for, respectively. Vector overhangs were introduced by placing primers up- (L-46nt-before-FseI) and downstream (L-48nt-after-SfoI) of restriction enzyme recognition sites. VSV-P-Lprot was cloned without (GGSG)$_3$ linkers in L to restrict spatial movement of the protease dimer and avoid potential interaction with the linked-dimer in P. Fragments 2 and 3 in this variant therefore received overhangs to the protease cleavage site at the 5′ end with primer L-1620-prot-rev and 3′ end with primer L-1620-prot-for, respectively. The protease was amplified with primer pair prot-for / prot-rev. The OFF switch constructs were generated as follows. The VSV-GFP plasmid was digested with MscI (inside GFP) and HpaI (inside L) to remove the intergenic region between GFP and L. Three overlapping fragments were generated by PCR to replace the C-terminal part of GFP (fragment 1: primers GFP-33nt-before-MscI-for to GFP-GGSG-rev or GFP-cut1-rev), to insert the HIV protease dimer (fragment 2: primers (GGSG)$_3$-prot-for to (GGSG)$_3$-prot-rev and prot-for to prot-rev) and to replace the N-terminal part of L (fragment 3: GGSG-L-for or cut2-L-for to L-N-term-rev). Fragment 1 had an overlap to the vector VSV-GFP and an overlap to the N-terminus of the protease insert, either with a (GGSG)$_3$ linker or without a linker sequence. Fragment 2 was either an HIV protease dimer with or without (GGSG)$_3$ linker. Fragment 3 had an overlap with the C-terminus of the protease insert and a 30 bp overlap to L. The three fragments were first joined by a fusion PCR. The resulting fusion fragment was cloned into the digested vector by Gibson assembly. In vitro comparison of both flexible and rigid Prot-OFF viruses revealed that the construct without linker was more sensitive to SQV control and hence was chosen for subsequent in vivo experiments. For sequence confirmation, viral RNA was purified by Viral DNA/RNA Kit, peqGOLD (Peqlab/VWR, Darmstadt, Germany). Subsequently, cDNA synthesis was performed with RevertAid First Strand cDNA Synthesis Kit (ThermoFisher, Vienna, Austria). PCR was performed with Q5® Hot Start High-Fidelity DNA Polymerase (NEB).

VSV-DsRed-ΔP (recombinant VSV Indiana strain lacking the viral envelope protein P) was generated by exchange of P with DsRed via pDsRed-Express-N1 (Clontech)[23]. In brief, a VSV Indiana plasmid was digested with BstZ17I and MluI. C-terminal N after BstZ17I and the intergenic region after N were PCR amplified by primer pair XhoI-N-for/EcoRI-N-rev and digested with XhoI and EcoRI. The intergenic region before and after M as well as M itself were PCR amplified by primer pair NotI-M-for/SacII-M-rev and digested with NotI and SacII. DsRed was subcloned from pDsRed-Express-N1 into the multiple cloning site of the pBluescript-II cloning vector (Stratagene) with BamHI/NotI. Digested C-terminal N and M PCRs were sequentially cloned into the pBluescript-II cloning vector before and after DsRed. The whole cassette was excised with BstZ17I and MluI and ligated into the digested VSV Indiana plasmid. Infectious viruses were retrieved in the presence of 10 μM amprenavir (THP Medical Products, Vienna, Austria) for prot-ON viruses and without amprenavir for non-protease containing viruses and prot-OFF viruses. Viruses were rescued using a helper virus-free calcium phosphate technique in 293T cells[53]. In brief, cells were transfected with expression plasmids of T7 polymerase (10 μg), VSV N (2.8 μg), P (1.8 μg), and L (0.6 μg) together with the desired vector (10 μg). To facilitate transfection, 25 μM chloroquine was added to the medium. In total, 2–6 days after transfection, cells were detached and transferred to BHK-21 or 293-VSV cells and cultured until clear cytopathic effect was observed. Virus was subsequently passaged on suitable cells 1-3 times. Finally, virus was plaque-purified twice and amplified on BHK-21 cells. Virus supernatants were filtered with 0.45 μm filter and concentrated through sucrose cushion (20%) via low-speed over-night centrifugation. Virus pellet was resuspended in PBS, aliquoted, stored at −80 °C, and titrated with TCID$_{50}$ assay. Structure visualization and molecular modeling were performed using Coot 0.8.7.1[54] and UCSF Chimera 1.12[55]. Images of molecular structures were generated with UCSF Chimera 1.12. Pprot structure models were generated with I-TASSER 5.1 standalone version with settings LBS = true (not depicted in model structure), light = true, hours = 6. Template libraries were retrieved from the I-TASSER server as of Jan 2019[56].

**Cell lines.** BHK-21 cells (American Type Culture Collection, Manassas, VA) were cultured in Glasgow minimum essential medium (GMEM) (Lonza, Basel, Switzerland) supplemented with 10% fetal calf serum (FCS), 5% tryptose phosphate broth, and 100 units/ml penicillin and 0.1 mg/ml streptomycin (P/S) (Gibco, Carlsbad, California, USA). In all, 293T cells (ATCC, used to generate 293tsA1609neo) and 293-VSV (293 expressing N, P-GFP and L of VSV)[57] and U87 cells (ATCC) were cultured in Dulbecco's Modified Eagle Medium (DMEM) supplemented with 10% FCS, P/S, 2% glutamine, 1x sodium pyruvate, and 1x non-essential amino acids (Gibco). G62 cells (kindly provided by M. Westphal; University Hospital Eppendorf, Hamburg, Germany) were cultured in DMEM supplemented with 10% FCS and 2% glutamine and P/S.

**Antibodies.** Primary antibody α-VSV-N (10G4, #EB009, raised in mouse, Kerafast, Inc., Boston, USA) Secondary antibody goat-α-mouse IgG2a Alexa 594 (A-21135, ThermoFisher, Vienna, Austria).

**In vitro experiments.** For compound dose responses, viral progenies were collected 24 hpi after an infection with an MOI of 1 from BHK-21, supplied with standard medium containing a range of PI concentrations. For replication kinetics, single step growth curves (MOI 3) were performed on BHK-21 cells with 10 μM

APV (Prot-ON constructs) or without PI (Prot-OFF). Samples were collected at indicated time points. Virus titers were determined using a 50% tissue culture infective dose (TCID50) assay using the method of Spearman–Kärber[58]. Fluorescent reporter gene expression was performed using a fluorescence microscope (Eclipse Ti, Nikon CEE GmbH, Vienna, Austria) with NIS-Elements BR 4.20.01 software (Nikon). For high magnification and live cell imaging of subcellular GFP signals, a Zeiss Axiovert 200 M microscopy was used controlled by VisiView (4.1.0.3.) software (Visitron Systems, Germany).

For testing genetic stability of the ON switch systems, serial virus passage on BHK-21 cells was performed in optimal (10 μM) and suboptimal (1 μM) amprenavir concentrations. After each passage protease inhibitor dependency was assessed by GFP expression after transferring a sample of the supernatant onto parallel dishes incubated without amprenavir. After 20 passages (P20), viral genomic RNA collected from the suboptimal amprenavir-treated virus propagation was purified, reverse transcribed, and a PCR performed on the inserts P-196PR2 and L-1620PR2. Fragments of 1491 and 1831 base pairs encoding the linked dimer and adjacent P and L sequences were generated by the primers pairs P-for/P-rev and L-for/L-rev with proofreading NEB Q5 polymerase. A VSV variant without protease insertion was used as negative control (P: 774 bp, L: 1114 bp). Prot-OFF virus was serially passaged 20 times in the absence of protease inhibitor and functionally tested afterwards with saquinavir. RNA purification, cDNA synthesis, PCR using primers GFP-33nt-before-MscI-for, and L-N-term-rev generating a fragment 1487 base pairs from VSV-Prot-OFF-GFP and 824 base pairs from VSV-GFP were performed. PCR fragments were sequenced by Sanger sequencing.

For plaque assay, monolayers of BHK-21 cells were infected with serial dilutions of virus stock in the presence or absence of PI. One hour after infection, cells were washed twice with PBS and overlayed with a 1:1 dilution of 2.5% plaque agarose and complete GMEM medium (including ±PI). The following day the plaque agarose was removed and cells were stained using crystal violet.

Samples for immunoblotting were collected from 293-VSV cells expressing VSV-N, P-GFP, and L[57]. We used this replication-supporting cell line for the expression of sufficient HIV protease dimer (Prot-ON - PI) and fusion protein (Prot-OFF + PI) in the non-active condition of respective viruses. SDS-PAGEs of protein lysates were performed under reducing conditions on a 8% polyacrylamide gel for the larger fusion proteins (+ PI) and 12% for the cleaved protease (− PI). Low percent gels were run for 60 min, high percent gels for 80 min. Proteins were transferred to 0.45-μm nitrocellulose membranes (Whatman, Dassel, Germany) by using a tank blotting system. The blotting time was 80 min for larger fusion proteins (+ PI) samples and 40 min for cleaved HIV protease dimer (− PI) samples. Blotting buffer contained 10% methanol for larger fusion proteins (+ PI) samples and 20% for cleaved HIV protease dimer (− PI) samples to reduce membrane pore size. Seperate gels for cleaved protease and uncleaved fusion proteins were used to meet optimal conditions for both sample types (i.e. small gel pores and long blotting time for fusion proteins; large gel pores and short blotting time for cleaved protease). The membranes were blocked over night with 1x PBS containing 5% skim milk and 0.1% Tween 20 (PBSTM). HIV protease dimer was stained by an antiserum[19] diluted 1:1000.

**Animal studies.** Animal experiments were approved by the Institutional Animal Care and Use Committee (ZVTA) from the Medical University Innsbruck and the Austrian Ministry of Science (BMBWF) in accordance with the "Tierversuchsgesetz 2012" (BGBI, I Nr 114/2012). Six- to eight-week-old female NOD.CB-17-Prkdcscid/Rj and athymic nude mice were purchased from Janvier Labs (Le Genest-Saint-Isle, France) and housed in a BL2 facility with a 12-h light/dark cycle with unrestricted access to food and water. Temperature in animal facilities was 20–24 °C for NOD.CB-17-Prkdcscid/Rj and 22–26 °C for athymic nude mice, humidity was 55 ± 10%. Mice were acclimated for one week prior to start of experiments. Statistical methods were not employed to predetermine sample size, numbers were chosen based on previous studies published by others and us. In vivo studies were performed unblinded. PI treatment commenced at the indicated time points. Amprenavir APV (0.8 mM; every 12 h) and saquinavir SQV (0.8 mM; every 8 h) where combined with ritonavir RTV (0.2 mM) in a 4:1 ratio in a buffer containing 10% DMSO, 40% PEG300, 5% Tween80 and 45% PBS, and applied intraperitoneally in 50 μl. For neurotoxicity assessment, BALB/c mice were anesthetized with ketamine and xylazine (100 and 10 mg/kg body weight, respectively) and stereotactically injected into the right striatum (2 mm and 0.4 mm rostral to bregma; depth at 3 mm) with 2 μl virus solution containing $2 \times 10^5$ TCID$_{50}$ VSV variants in PBS. PI treatment (APV + RTV, every 12 h) was initiated one hour before virus application. High-dose indinavir sulfate (100 μl per mouse of 9 mM + 1 mM RTV in 10% DMSO, 40% PEG300, 5% Tween80, and 45% PBS) was carried out every 6 h due to shorter half-life of the compound. Mice were monitored twice daily with weight measurements and clinical toxicity score assessment that included neurological signs (paralysis, coordination, motility, grasp reflex) and indications for general well-being (body condition, grooming, posture, and spontaneous movements). For subcutaneous xenografts, 100 μl glioblastoma cell suspension containing $2 \times 10^6$ human U87 or G62 glioma cells were injected into the right flanks of NMRI nude mice (Janvier) or NOD-SCID mice (Janvier), respectively. To test the ON-switch system, U87 xenografts grown in athymic nude mice with a median volume of 0.1 cm$^3$ were intratumorally injected with a single maximum feasible dose of 30 μl containing $2 \times 10^6$ virus TCID$_{50}$ of the indicated VSV variants

or control buffer. PI treatment (APV + RTV) was initiated 1 h before virus application. To test the OFF-switch system, G62 xenografts in NOD/SCID mice with a median volume of 0.07 cm$^3$ were intratumorally injected with 30 μl containing the maximum feasible dose of $2 \times 10^7$ virus TCID$_{50}$ of the indicated VSV variants or control buffer. Virus treatment was repeated seven days later. PI treatment (SQV + RTV) was initiated either 8 days post second virus injection for studies on tumor control or 3 days post single virus treatment for histological studies. Bioluminescence in vivo imaging of luciferase-expressing VSV variants was performed with an IVIS® Lumina II (Perkin Elmer, Waltham, MA) system as previously described[59]. In brief, NMRI nude mice (Janvier) bearing subcutaneous U87 tumors were treated intratumorally with $2 \times 10^6$ (TCID50) of luciferase-expressing VSV-Pprot-Luc. In all, 1.5 mg D-luciferin (Promega, Madison, WI, USA) was administered intraperitoneally 15 min prior to measurements. Luminescence data acquisition and analysis was performed using Caliper Live Sciences-Living Image® software (4.3.1.). Tumors were measured with a caliper and volume was calculated using the formula: length × width$^2$ × 0.4. Humane endpoints were defined by either tumor size >1.5 cm$^3$, weight loss >20% or clinical signs of neurotoxicity. Mice were euthanized via cervical dislocation after isoflurane anesthesia. For histological analysis, VSV variant-treated tumors were harvested 10 days post inoculation and fixed in 4% PFA at 4 °C over night, dehydrated in 30% sucrose, and sliced at −20 °C into 7-μm-thick sections (Microm HM560, ThermoFisher Scientific). Slices were blocked with 2% normal goat serum (G9023, Sigma, Vienna, Austria), 2% BSA (Roth, Karlsruhe, Germany) and 0.01% Triton-X. Primary antibody α-VSV-N (10G4, #EB009, raised in mouse, Kerafast, Inc., Boston, USA) was diluted 1:250 and incubated over night. Secondary antibody goat-α-mouse IgG2a Alexa 594 (A-21135, ThermoFisher, Vienna, Austria) was diluted 1:750 and incubated for 30 min. Histological sections were analyzed using a fluorescence microscope (Eclipse Ti, Nikon CEE GmbH, Vienna, Austria).

**Visual presentation**. Contrast and color of the photomicrographs were linearly adjusted with Adobe Photoshop. Cartoons and schematic illustrations were generated with BioRender (biorender.com), Adobe Photoshop, and Inkscape 0.92.3.

**Statistical analysis**. GraphPad prism software (GraphPad Software, Inc., La Jolla, CA) was used for statistical analysis and data presentation. Statistical significance was determined by Student's $t$ test and analysis of variance (ANOVA) with Turkey's multiple comparison correction. $P$ values of <0.05 were considered statistically significant. Kaplan–Meier survival curves were compared using the Log-rank (Mantel-Cox) test. Statistically significant differences were indicated as follows: *$p < 0.05$; **$p < 0.01$; ***$p < 0.001$.

**Reporting summary**. Further information on research design is available in the Nature Research Reporting Summary linked to this article.

## Data availability

All pertinent data to support this study are included in the manuscript and supplementary files. Source data are provided with this paper. Further data supporting the findings are available upon request. The sequences of the prot constructs have been deposited at NCBI's GenBank with the accession codes MW316665 (not codon optimized) and MW316666 (codon optimized). Recombinant virus variants described herein can be made available with a Material Transfer Agreement (MTA). Requests require the review and approval of ViraTherapeutics GmbH to confirm compliance with intellectual property and confidentiality obligations. Source data are provided with this paper.

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

## Acknowledgements

We thank J. Rinnofner, M. Salvatore, and L. Riepler for excellent technical support and A. Volland for helpful discussions and S. Geley for providing high-resolution life cell imaging equipment. This project was supported by grants from the Austrian Research Promotion Society FFG (program "Bridge" # 4766773) and European Union's Horizon 2020 research and innovation program (# 681032, "European HIV Vaccine Alliance") to D.v.L and J.K. and a grant by the Christian Doppler Research Association to G.W.; E.H. has been a recipient of a DOC Fellowship of the Austrian Acadamy of Science.

## Author contributions

D.v.L. conceived the initial concept. E.H., G.W., and D.v.L. designed the experiments. E.H., J.K., L.E., and T.N. conceived cloning strategies. H.G.K. assisted in protease construct design and PI application strategy. E.H. and B.H. generated recombinant viruses. E.H., B.H., A.R., I.B., C.U., and G.W. performed experiments. E.H., H.G.K., G.W., and D.v.L. wrote the manuscript. All authors read and approved the final manuscript.

## Competing interests

A patent application relating to all aspects of the manuscript has been filed under the application number 19181717 by Boehringer Ingelheim International GmbH (application date 21 June 2019). E.H., J.K., B.H., L.E., T.N., G.W., and D.v.L. are listed as inventors. T.N., C.U., and L.E. are employed by ViratTherapeutics GmbH. D.v.L. is founder of ViraTherapeutics GmbH. D.v.L. and G.W. serve as scientific advisors to Boehringer Ingelheim Pharma K.G. ViraTherapeutics or Boehringer Ingelheim had no role in the study design, data analysis and interpretation, or the writing of the manuscript. The other authors declare no competing interests.
