## [Peer Review File · Nature Communications]

Reviewers' Comments:

Reviewer #1:

Remarks to the Author:

Von Laer and co-workers describe a clever approach to control replication of engineered VSV by introducing HIV protease and respective cleavage sites into VSV essential proteins and to address the protease with clinically used protease inhibitors. Depending on the insertion sites, both ON- and OFF-switches can be generated. The system seems to function robustly, although not much effort is presented to even further optimize its efficiency. Nevertheless, I am convinced that this approach is very interesting for controlling RNA virus replication in some future applications. Some general issues should be addressed:

- The authors claim that the mechanism should be universally applicable to all kinds of RNA viruses. It would be interesting to get at least a more careful discussion of this issue, if not an actual demonstration of the concept in another RNA virus.
- The advantages of using the system compared to RNA-based systems for controlling viral replication have been pointed out. However, the authors should also mention some disadvantages of the present strategy such as the introduction of a further viral protein, the limitation to control replication instead of only transgene expression (which could limit its applicability), etc.
- In general, it would be more informative to characterize the presented in vitro systems mechanistically beyond the descriptive observation of viral replication and reporter expression. For example, could there be a demonstration of the regulation of protease activity on the protein level, e.g. by observing different protease cleavage activities by western blots of the respective viral proteins?
- The reduced neurotoxicity of the engineered VSV is impressive, however it is not immediately clear why this is observed. I.e. is the limited brain penetration of APV the main reason? Are there inhibitors that show better penetration in order to prove this hypothesis? This issue needs clarification.
- A recent paper by the Yokobayashi group describes the use of aptazymes controlling transgene expression in VSV (ACS Synth Biol. 2019, 8(9), 1976-1982. doi: 10.1021/acssynbio.9b00177), please cite and discuss.

Reviewer #2:

Remarks to the Author:

This is a short and relatively simple paper but it tests an elegant and potentially clinically useful concept of limiting the replication of therapeutic RNA virus by introducing elements which allow protease-dependent ON or OFF states. For the reason that the concept is largely sold as a clinically useful concept, I judge the paper on that basis as well as on the basis of the data that are shown.

Figure 1 b and d – the 'models' are not very sharp in their outlines and although its understood what they are meant to show, they do not really illuminate the text. The pictures of GFP+ cells probably ought to be stylised too, to prevent them pre-empting actual data

I'd like to see formal growth curves of the modified versus unmodified viruses in the constant presence of an adequate concentration of drug. From fig 1f it appears the titres are about 1 log lower than the unmodified viruses

The authors state that the virus was genetically stable after many rounds of passage in vitro. What

do they mean by genetically stable? Did they sequence the whole viral genome? Tested after being administered in vivo? This is important because if these viruses are used for clinical application with the expectation of being able to control them, I think the authors need to try a bit harder to clearly demonstrate to the reader that they do retain control of their viruses after various situations.

With both viruses, a fairly hefty dose of amprenavir is needed to get them up to a suitable titre. Can the authors please comment on the drug levels needed and the compatibility with attaining those drug levels in a sustained manner in humans? Given Amprenivir is no longer in production and has long been replaced by a prodrug fosamprenavir, the authors comment specifically on whether the levels are compatible with amprenavir levels achieved by prodrug administration.

This comment applies throughout, for all PIs used - please address if for all

For figure 3, a different murine model is used, a different cell line and different dose of virus, can the authors explain why? It would seem better for the readers to compare the data if both models were used with both sets of viruses

As in figure 1, they should show formal viral growth curves for all viruses used – in this case of course without PI drugs

3d the grey and orange points curves only appear as one point – would be better to show separate graphs for the two different PI used

As a general comment about discussion - the authors have shown their system works for one virus, VSV but the discussion rather overreaches into a discussion of environmental spread . In Line 226-7 they state and an environmental spread is frequently observed for some life attenuated vaccines – statement needs referencing –it may be my ignorance but apart from no longer used live polio vaccine I am not aware of environmental spread of human vaccine viruses I think it is unlikely the risk:benefit assessment of vaccines will include all human vaccinees taking medications with their vaccines

If the authors are referring to animal viruses and vaccines they need to clarify and reference their statements

Although the introduction refers to the work of others, the discussion does not refer to or compare their system with all the other systems which have been developed to mitigate the neurotoxicity of VSV

The final conclusion "Taken together, the Prot-OFF and Prot-ON systems presented here allow effective control of therapeutic and vaccine viruses " is a total overreach. The authors have shown data for one oncolytic virus only. They may imply that this is generalizable phenomenon but that implication is, of course, a testable and it has not been tested by the authors using any other viruses.

Minor/typos

Line 255 live not life

Line 509 cocktail not ocktail

Referee response letter

We would like to thank the reviewers for the very helpful and constructive suggestions. We have thoroughly revised the manuscript and added a substantial amount of new data. A point-by-point response to all of the referee's comments is listed below.

The response letter has been structured the following way:

Reviewers comments in grey

Responses in indented text

Revised text passages in italics

Reviewer #1 (Remarks to the Author):

Von Laer and co-workers describe a clever approach to control replication of engineered VSV by introducing HIV protease and respective cleavage sites into VSV essential proteins and to address the protease with clinically used protease inhibitors. Depending on the insertion sites, both ON- and OFF-switches can be generated. The system seems to function robustly, although not much effort is presented to even further optimize its efficiency. Nevertheless, I am convinced that this approach is very interesting for controlling RNA virus replication in some future applications. Some general issues should be addressed:

We appreciate the positive and supportive notion of this reviewer and feel confident that this novel regulation principle for RNA virus activity will be further validated in future studies. Additional studies were performed when feasible to address some of the comments. We have addressed all comments and suggestions of this reviewer as outlined below:

The authors claim that the mechanism should be universally applicable to all kinds of RNA viruses. It would be interesting to get at least a more careful discussion of this issue, if not an actual demonstration of the concept in another RNA virus.

Following this suggestion, we have added a more detailed discussion of the possibility to expand the described system to other viruses; the added text is given below. We believe that adapting the Prot-ON approach to other viruses should be rather straightforward, while the applicability of the Prot-OFF principle may be less easy to predict. We are confident that the described principles will be taken up by other groups working on the respective viruses, while adapting the principle of conditional proteolytic control to a completely different system would have been beyond the scope of the current study.

Discussion page 13-14; lines 336-353

Although not the focus of the current study, we hypothesize that the presented regulatory principles can generalize to viruses other than VSV if certain conditions are met. As a central precondition to generate Prot-ON RNA viruses, target genes should be an essential part of the viral replication complex and be permissive for

intramolecular insertions of transgenes. Such insertion sites have been described for several viruses, e.g. measles virus⁴³, rinderpest virus⁴⁴, canine distemper virus⁴⁵, Ebola virus⁴⁶ or rabies virus⁴⁷. Insertion herein of the HIV protease, or other proteases amenable to autoproteolytic activity, should generate a regulatable Prot-ON virus similar to the presented VSV-Pprot. Conversely, the Prot-OFF principle of virus activity control should also work for other viruses when two stipulations are met. For one, replication-essential genes need to tolerate minor N- or C-terminal tags comprised of the residual amino acids of the protease recognition sites after cleavage. However, without such proteolytic cleavage these gene products would have to remain non-functional fusion constructs in the presence of large tags such as fluorescence proteins^{48, 49}. While numerous fluorescent protein fusion tags have been described on viral structural proteins^{49, 50}, studies on terminal fusions to proteins involved in viral replication are limited⁵¹. Here, additional studies might be necessary to evaluate a virus' suitability for the OFF switch.

The advantages of using the system compared to RNA-based systems for controlling viral replication have been pointed out. However, the authors should also mention some disadvantages of the present strategy such as the introduction of a further viral protein, the limitation to control replication instead of only transgene expression (which could limit its applicability), etc.

We have added some considerations regarding the shortcomings of our conditional proteolysis approach to the discussion. These included also thoughts on a direct comparison with the aptazyme approach (also below response to suggestion on discussion of ref. Takahashi and Yokobayashi, 2019).

Discussions Page 11; lines 275-285

Our presented ON and OFF switch system focuses on viral replication control and therefore transgene expression levels are linked to viral replication. Conversely, the guanine aptazyme approach enables the control of transgenes independent from the virus, which might present an advantage over our system in certain applications. Hypothetically, the combination of RNA- and protein-based regulation approaches could hold potential for future developments of regulatable viruses, e.g. to regulate virus replication and transgene expression by different small molecules or to have multiple, completely different safety mechanisms in place as fail safe. Lastly, another advantage of RNA-based regulation via aptazymes is the independence from additional xenogenic proteins within the vector, which can be immunogenic or attenuate the virus.

In general, it would be more informative to characterize the presented in vitro systems mechanistically beyond the descriptive observation of viral replication and reporter expression. For example, could there be a demonstration of the regulation of protease activity on the protein level, e.g. by observing different protease cleavage activities by western blots of the respective viral proteins?

We followed this suggestion and performed Western blots for switch constructs VSV-Pprot-GFP, VSV-Lprot-GFP and VSV-Prot-OFF. Antibodies against VSV P or L are not commercially available; we therefore targeted the HIV protease with a rabbit polyclonal antiserum previously shown to bind the protease both in its

monomeric and dimeric form (Kraeusslich, 1991, PNAS). The Western blots directly confirm conditional autoproteolytic cleavage depending on PI presence for both, the ON and OFF switch design, respectively. The Western blot data have been included as **new Figure 1c, 1f, and 3c**.

Results Page 4-5 ; lines 104-107

Western blotting against HIV protease confirmed the proteolytic cleavage in the absence of PI with a band marking the HIV protease dimer at about 22 kDa¹⁹, compared to a band at 54 kDa corresponding to the Pprot fusion protein in the presence of PI (Figure 1c).

Results Page 5 ; lines 123-126

Consequently, Western blot analysis revealed an intact large L protein HIV protease dimer fusion protein of about 266 kDa in the presence of PI and a separated HIV protease dimer band of 22 kDa in the absence of PI (Figure 1 e,f).

Results Page 8 ; lines 199-202

Western blotting against HIV protease displayed a large fusion protein, matching the combined size of the VSV polymerase, GFP, and the HIV protease dimer (~290 kDa) (Figure 3c).

The reduced neurotoxicity of the engineered VSV is impressive, however it is not immediately clear why this is observed. I.e. is the limited brain penetration of APV the main reason? Are there inhibitors that show better penetration in order to prove this hypothesis? This issue needs clarification.

The reviewer highlights a very important issue. In general, the established HIV PIs feature limited CNS access. Among the PIs, amprenavir, the compound used in the majority of our experiments, is at the lower end of CNS penetration. Therefore, it actually constitutes an additional layer of safety, as systemic PI levels sufficient for peripheral regulation of VSV-prot-ON switches would not support intracranial virus activity, and therefore abrogate the risk of neurotoxicity. However, as there are PI's in use with better intracranial availability (Ene et al, J Med Life, 2011), we performed an additional in vivo experiment aiming for enhanced PI presence during and after the stereotactic procedure. Therefore we tested if by adding PI directly into the injectate and by using indinavir, one of the PIs with highest CNS penetrability, we could detect signs of neurotoxicity or intracranial spread. While the parental VSV-GFP showed rapid neurotoxicity, neither of the mice treated with any of the enhanced PI regimen showed any. Histological analysis of the brains also did not reveal any spread beyond the immediate lining of the injection track. We have added new **figures 2i, j and S7c** to the results and a new part in the discussion.

Results Page 7-8 ; lines 181-189

As the CNS penetrability of APV is low²⁷, we performed a complementary neurotoxicity study with the VSV-Lprot-GFP virus applying an alternative PI treatment regimen with indinavir (combined with RTV; this combination has been shown to have a higher CNS penetration²⁸) as well as providing 10 μ M APV directly in the stereotactic injection volume. Despite the initial presence of PI in

the injectate and the use of a PI regimen with higher expected CNS concentration, no signs of neurotoxicity or continued weight loss were observed (Figure 2i, S7c), and the virally induced GFP signal was still confined to the injection site (Figure 2j).

Discussion Page 12-13 ; lines 308-319

The most consistent way to trigger neurotoxicity in mice, is the direct intracranial instillation of VSV. With our conditional proteolysis ON-switch approach, another factor comes into play, however. Most, if not all, HIV protease inhibitors show only limited penetration into the CNS²⁷, thus not allowing PI-dependent activation of viral replication in the CNS. Even applying a PI with higher CNS availability, and adding PI to the injectate, no neurotoxicity and viral spread could be observed in our experiments. We believe this actually adds an additional layer of safety to the system, as the PIs required to maintain activity of the ON switch constructs and thus mediate neurotoxicity only reach a fraction of their serum levels in the central nervous system²⁷. This might be of importance for attenuated VSV variants, that are safe with peripheral application but have shown toxicity potential once entered into the brain⁴².

A recent paper by the Yokobayashi group describes the use of aptazymes controlling transgene expression in VSV (ACS Synth Biol. 2019, 8(9), 1976-1982. doi: 10.1021/acssynbio.9b00177), please cite and discuss.

This pertinent reference has been included in the revised manuscript. It has been added to the introduction and discussion (see also response to first question by this reviewer).

Introduction Page 2 ; lines 52-53

Of those, RNA-aptazymes fused to a viral gene were shown to regulate virus replication over a range of less than 100-fold⁷ and viral transgene expression up to 30-fold⁸.

Discussion Page 10-11 ; lines 259-285

Further development of aptazyme-controlled virus and transgene activity comprised a guanine-responsive switch in 3' UTRs of glycoprotein deficient VSV expressing GFP. The strongest transgene repression of 26.8-fold was facilitated by placement of the switch in both 3' UTRs of GFP and the VSV polymerase in the presence of 500 μ M guanine⁸. However, as discussed therein guanine's low solubility impedes in vivo applications. Our system, on the other hand, is based on clinically approved and extensively deployed PIs with potent control of viral replication in the low micromolar range. Such doses are comparable to dose equivalents of oral treatment regimens in human for amprenavir, saquinavir or indinavir, respectively^{30, 31} or precursors³², respectively. Arguably, as the potential control of therapeutic viruses encompasses a much shorter duration compared to the chronic application of HIV PIs, the application of even higher doses of PIs might be conceivable. Of note, our in vitro dose response studies revealed that higher doses of PI are required compared to their reported EC50 concentrations for in vitro control of HIV¹⁸. However, the APV and SQV doses applied intraperitoneally to study controllability in vivo studies were in line with

reported human plasma levels under standard therapy^{33, 34}. Our presented ON and OFF switch system focuses on viral replication control and therefore transgene expression levels are linked to viral replication. Conversely, the guanine aptazyme approach enables the control of transgenes independent from the virus, which might present an advantage over our system in certain applications. Hypothetically, the combination of RNA- and protein-based regulation approaches could hold potential for future developments of regulatable viruses, e.g. to regulate virus replication and transgene expression by different small molecules or to have multiple, completely different safety mechanisms in place as fail safe. Lastly, another advantage of RNA-based regulation via aptazymes is the independence from additional xenogenic proteins within the vector, which can be immunogenic or attenuate the virus.

Reviewer #2 (Remarks to the Author):

This is a short and relatively simple paper but it tests an elegant and potentially clinically useful concept of limiting the replication of therapeutic RNA virus by introducing elements which allow protease-dependent ON or OFF states. For the reason that the concept is largely sold as a clinically useful concept, I judge the paper on that basis as well as on the basis of the data that are shown.

We appreciate the constructive feedback and suggestions to clarify some incongruences and modify some of our interpretations. Additional studies were performed when feasible to address some of the comments. We have addressed all points of this reviewer as outlined below:

Figure 1 b and d – the ‘models’ are not very sharp in their outlines and although its understood what they are meant to show, they do not really illuminate the text. The pictures of GFP+ cells probably ought to be stylised too, to prevent them pre-empting actual data

We agree with the reviewer’s suggestions and have revised the corresponding figures accordingly:

The schematic cartoons in **new Figure 1b** and e as well as in **new Figure 3b** were redrawn to highlight the functional principle.

The GFP images in the schematic section of Fig 1 were removed.

We added a **new figure 1g** showing exemplary GFP expression photomicrographs from a protease inhibitor dose response study.

For Figure 3, we changed the order of the GFP fluorescence photomicrographs and placed them as a part of the results presentation as **new Fig 3d**.

I’d like to see formal growth curves of the modified versus unmodified viruses in the constant presence of an adequate concentration of drug. From fig 1f it appears the titres are about 1 log lower than the unmodified viruses

We agree that formal single step growth curves add to the characterization of the regulatable constructs. To address this point, growth curves for VSV-Pprot-GFP, VSV-Lprot-GFP (**new Figure 1j**) and VSV-Prot-OFF-GFP (**new Figure 3f**) as well as for the parental virus VSV-GFP were generated. Of the three modified constructs, VSV-Pprot-GFP showed an attenuated viral replication in the range of 1-1.5 log, while the others exhibited similar replication as VSV-GFP.

Results Page 5-6 ; lines 128-133

We next compared the single step growth curve kinetics of the two regulatable virus variants with the parental VSV-GFP. In the presence of 10 μ M APV VSV-Lprot-GFP showed no significant attenuation in replication kinetic while VSV-Pprot-GFP showed mild attenuation in the range of 1-1.5 logs compared to parental viral titers (Figure 1j).

Results Page 8 ; lines 208-210

Single step replication growth kinetics revealed no attenuation of the Prot-OFF construct compared to the parental VSV-GFP (Figure 3f).

The authors state that the virus was genetically stable after many rounds of passage in vitro. What do they mean by genetically stable? Did they sequence the whole viral genome? Tested after being administered in vivo? This is important because if these viruses are used for clinical application with the expectation of being able to control them, I think the authors need to try a bit harder to clearly demonstrate to the reader that they do retain control of their viruses after various situations.

We thank the reviewer for pointing this out and we try to clarify this issue in the revised version of the manuscript. Our statement regarding stability concerned the continued PI dependence as well as genetic stability of the insert over multiple passages in vitro and when being re-isolated from tumors. We did not, however, sequence the entire genome of VSV after multiple passages or re-isolation from in vivo applications, and this is now also stated in the Results. Clearly, no second site mutations occurred that would have affected the biological properties of the virus, while we cannot exclude that mutations outside the insertion region may have occurred.

We now added details to the corresponding results chapters and outlined our approach in the methods. We also generated **new Figures S3, S5, and S9** that each contains 4 subfigures addressing different aspects of our approach. Throughout the in vitro serial passage, we tested newly generated progeny for their PI dependence (for the ON switch constructs), we used an insert-spanning PCR to evaluate potential loss of regulatory inserts and we sequenced the regulatory construct after 20 passages. Using these multiple measures, we did not find an indication for loss of PI control after up to 20 passages. In addition, we also tested ON-switch constructs after in vivo passage by isolating virus from tumor lysate and assessing PI dependency in vitro.

Results Page 5 ; lines 107-118

To address genetic stability, we performed in vitro serial virus passages in optimal (10 μ M) and suboptimal (1 μ M) amprenavir conditions. After each passage protease inhibitor dependency was assessed by GFP expression in the presence or absence of PI (Figure S3a). After 20 passages (P20), no amprenavir escape virus variants could be observed and functional PI dependency was confirmed by GFP fluorescence and viral plaque assay (Figure S3b). A protease-insert-spanning PCR was used to confirm the presence of a single band matching the size for an intact PR dimer insert (Figure S3c). Subsequent sequencing of the P-prot insert (not the whole viral genome) and alignment comparison with the

parental plasmid construct revealed one mutation (protease 2: nucleotide G23A; amino acid R8K) in the construct with no effect on the proteolytic control of virus activity (Figure S3d).

Results Page 6 ; lines 133-136

Analogous to the Pprot variant above, twentyfold serial passage in the presence of both optimal (10 μ M) and sub-optimal concentrations (1 μ M) of APV did not produce any viral progeny lacking dependency on protease inhibition (Figure S5).

Results Page 7 ; lines 163-167

Importantly, virus isolated from select tumors maintained PI dependency in vitro and sequencing of the insert region of either VSV-Pprot-Luc or VSV-Lprot-GFP virus isolates after in vivo passage revealed no change in sequence compared to the start virus, highlighting the robustness of the regulatable construct.

Results Page 9 ; lines 215-217

Consistent with the observed genetic stability with the ON switch constructs, repetitive in vitro passage over 20 cycles did not change the responsiveness of the OFF switch (Figure S9).

With both viruses, a fairly hefty dose of amprenavir is needed to get them up to a suitable titre. Can the authors please comment on the drug levels needed and the compatibility with attaining those drug levels in a sustained manner in humans? Given Amprenavir is no longer in production and has long been replaced by a prodrug fosamprenavir, the authors comment specifically on whether the levels are compatible with amprenavir levels achieved by prodrug administration.

This comment applies throughout, for all PIs used - please address if for all

Compared to the EC50 concentrations published for in vitro HIV control (Lv et al., 2015 HIV AIDS), which range from 6nM (IDV) and 38nM (SQV) to 12-80nM for APV, the concentration required to regulate the activity of our VSV constructs was 10-100 times higher. While this indicates a lower efficacy of PIs at controlling a conditional proteolysis RNA virus construct compared to the primary target HIV, the important question – as brought up by the reviewer – is if the doses required for in vivo activity can be reached within clinically approved levels.

Below we compare the applied in vivo doses for our studies and compare them with reported plasma levels in human under standard therapy. For APV and SQV the doses injected are below human plasma levels. IPV was used in our additional brain application study and we chose on purpose the maximum feasible dose as the treatment would be very short-term (10 days). This applied dose equals about 3 times the max plasma levels reported in human and we assumed this would be considered tolerable, particularly as the PI application is a temporary measure, unlike a chronic HIV treatment regimen.

Although APV is not a clinically applied drug anymore, we chose at the start of our studies as it is a classical PI drug with established in vitro activity. Fosamprenavir as a prodrug on the other hand shows little direct PI activity in vitro in the absence of drug conversion mechanisms associated with intestinal mucosa cells (Brouwers et al, 2006, Int J Pharmaceutics). We did compare a

number of PIs and included a new PI (indinavir, IDV) in our extension study of stereotactic brain injection and the monitoring of potential neurotoxicity. As our in vitro PI dose response for the VSV-prot-OFF construct showed higher potency of saquinavir compared to amprenavir, we continued our in vivo study with the first.

Study treatment	Mouse		Human		Reference
	Drug dose per mouse	Drug dose per weight	Human dose applied	Reported plasma levels	
	µg per mouse (30g)	µg per kg µg per ml	mg	µg/ml	
Amprenavir 0.8 mM ip 50 ul	20.2 µg	0.67 mg/kg 0.67 µg/ml	1200 b.i.d.	9.1 µg/ml	Sadler et al., 1999, Antimicrob Agents Chemother
Saquinavir 0.8 mM ip 50 ul	27.9 µg	0.92 mg/kg 0.92 µg/ml	1000 b.i.d.	2.1 µg/ml	Durant et al, 2000, AIDS
Indinavir 9mM 100ul	552.4 µg	18.2 mg/kg 18.2 µg/ml	800 t.i.d.	7.7 µg/ml	

We have added this information to the discussion.

Discussion Page 11 ; lines 271-275

Of note, our in vitro dose response studies revealed that higher doses of PI are required compared to their reported EC50 concentrations for in vitro control of HIV¹⁸. However, the APV and SQV doses applied intraperitoneally to study controllability in vivo studies were in line with reported human plasma levels under standard therapy^{33, 34}.

For figure 3, a different murine model is used, a different cell line and different dose of virus, can the authors explain why? It would seem better for the readers to compare the data if both models were used with both sets of viruses

In the revised version, we have clarified the rationale for choosing a different mouse model. For the subcutaneous tumor models, we first used the established U87 glioma model in flank tumor setting in athymic nude mice. This model has been established in our lab as a reproducible model for IVIS bioluminescence luciferase imaging. However, as athymic mice still contain functional NK cells as well as B cells, peripheral application of VSV does not lead consistently to neurotoxicity (previous observations in our lab). As the ability to control and prevent neurotoxicity is the central key feature of the OFF switch, we needed to employ a model more sensitive to peripheral VSV treatment. Hence we switched to NOD-SCID mice and the G62 glioma model; in a previous study this model showed consistent wildtype-VSV induced neurotoxicity after intratumoral injection (Muik et al, Cancer Research 2014). We added a corresponding explanation to the discussion. As for the dose differences, production of VSV-Prot-OFF-GFP yielded about a magnitude higher stock titers. As induction of toxicity (or better:

the prevention of such) was the primary focus of the Figure 3 in vivo xenograft study, we went with the maximum feasible dose of 2×10^7 , which was higher than the maximum feasible dose for the Prot-ON constructs used in the xenograft studies in Figure 2 (2×10^6). We have added this information to the method description as well.

Discussion Page 12 ; lines 298-308

Therefore, to address potential toxicities of replication-competent viral therapeutics in principle and to test our regulatable ON and OFF switch constructs in vivo, we had to focus on the wild-type VSV backbone, which reproducibly elicits neurotoxicity. However, application of wildtype-based VSV in the athymic nude mouse model did not elicit neurotoxicity, consistent with intact NK cell activity in these animals and corresponding incomplete immune deficiency. We therefore employed an alternative xenograft model for testing the ability to reduce neurotoxicity after activating the OFF switch in VSV-Prot-OFF-GFP: this made use of the highly immune deficient NOD-SCID host³⁸. In this model, we observed clear neurotoxicity, which could be successfully prevented when triggering the OFF switch as late as 15 days after the oncolytic treatment start.

As in figure 1, they should show formal viral growth curves for all viruses used – in this case of course without PI drugs

See comment above, **new Figure 3f**

3d the grey and orange points curves only appear as one point – would be better to show separate graphs for the two different PI used

Figure 3d was replaced by a **new Figure 3e and new suppl Figure S8a**. The initial figure showed a protease inhibitor dose response using amprenavir. For saquinavir, only one dose was tested. We now performed and present a full dose response with saquinavir instead. As we used saquinavir as the protease inhibitor to control VSV-Prot-OFF-GFP in the subsequent in vivo studies and did not perform any further studies with amprenavir, we exchanged the dose response in the main figure with SQV as the PI. The existing dose response with APV has been placed to the supplement figures.

Results Page 8 ; lines 206-208

Replication of this recombinant virus was inversely dependent on PI dose, and could be blocked by PIs SQV (Figure 3e) and APV (Figure S8a).

As a general general comment about discussion - the authors have shown their system works for one virus, VSV but the discussion rather overreaches into a discussion of environmental spread . In Line 226-7 they state and an environmental spread is frequently observed for some life attenuated vaccines – statement needs referencing –it may be my ignorance but apart from no longer used live polio vaccine I am not aware of environmental spread of human vaccine viruses

I think it is unlikely the risk:benefit assesement of vaccines will include all human vaccinees taking medications with their vaccines

If the authors are referring to animal viruses and vaccines they need to clarify and

reference their statements

Our study focuses on ON and OFF switches of VSV. We have added a designated paragraph to the discussion on the transfer of the principle of conditional proteolysis to other viral platforms and believe that future studies might present such developments (see comment to first point of Reviewer 1). But as such broad application has indeed not been shown yet, we markedly toned down the discussion on the additional environmental safety aspect and also added some explanation regarding the implication of exposure of VSV to livestock. We already had included a pertinent reference that discusses this issue based on a risk assessment of environmental spread of another oncolytic virus (SVV). We added a sentence describing this reference (Schijven et al, 2019, Risk Anal.)

Discussion Page 12 ; lines 286-294

In theory, the ON-switch system could also present as an additional environmental safety element, although not tested in this study. As virus progeny depend on the presence of PI, potentially shed virus is not active for productive infection of a new host. This could be of importance, when therapeutic or vaccine RNA viruses can cause or mimic notifiable animal diseases, as was recently discussed in a quantitative risk assessment estimate for oncolytic treatments with Seneca Valley virus (SVV)³⁵. Of note, while not being directly a serious pathogen for livestock, wild-type VSV infection clinically mimics foot-and-mouth disease, and therefore exposure to animals should be limited^{36 37}.

Although the introduction refers to the work of others, the discussion does not refer to or compare their system with all the other systems which have been developed to mitigate the neurotoxicity of VSV

We added a designated paragraph to the Discussion addressing VSV neurotoxicity, the choice of our models, the factor of protease inhibitor brain penetrability, and the choice of wildtype VSV over existing non-neurotoxic VSV variants. Our lab has engineered and developed a non-neurotoxic chimeric VSV variant, VSV-GP, with an excellent preclinical safety profile (Muik et al., 2014 Cancer Res.). If we had introduced our switch constructs into this chimeric platform, we would have been limited to study our constructs for therapeutic efficacy and reporter gene activity, but without assessing the potential to add a safety switch to a replicating biotherapeutic. Of note, one of the VSV variants (VSV- IFN- β) currently undergoing clinical testing and which presents with a convincing safety profile after peripheral application, was shown to still harbor neurotoxicity potential once it can enter the brain, for instance through CNS or meningeal metastases (Yarde et al., 2013 Cancer Gene Ther.).

Discussion Page 12 ; lines 289-295

Neurotoxicity of wild-type based VSV has been abrogated in recent years by genetic engineering for chimeric variants^{38, 39} or by naturally occurring relatives within the rhabdovirus family⁴⁰ with some of these variants having already entered clinical testing⁴¹. Therefore, to address potential toxicities of replication-competent viral therapeutics in principle and to test our regulatable ON and OFF switch

constructs in vivo, we had to focus on the wild-type VSV backbone, which reproducibly elicits neurotoxicity.

Discussion Page 13 ; lines 317-319

This might be of importance for attenuated VSV variants, that are safe with peripheral application but have shown toxicity potential once entered into the brain⁴².

The final conclusion “Taken together, the Prot-OFF and Prot-ON systems presented here allow effective control of therapeutic and vaccine viruses “ is a total overreach. The authors have shown data for one oncolytic virus only. They may imply that this is generalizable phenomenon but that implication is, of course, a testable and it has not been tested by the authors using any other viruses.

We believe the principle on our ON and OFF switch could generalize to other viruses and have added an extensive paragraph to the discussion on the prerequisites to be met in order to introduce the regulatory switch to other RNA virus platforms. See first reply to Reviewer 1. We hope that our study serves as proof of principle and groups with expertise in other RNA viruses might consider applying our switch principle on their respective platforms. We do, however, acknowledge that our final paragraph overstated our findings from a potential to a fact. We have therefore modified it accordingly.

Discussion Page 14 ; lines 354-359

Taken together, the Prot-OFF and Prot-ON systems presented here allow effective control of the oncolytic virus VSV in the host by several drugs that have a long-term clinical safety record. Our chemogenetic proteolysis approach might be used for other therapeutic or vaccine viruses as well. Thus, this technology can potentially support a broad range of applications and provides the basis for future developments of new safer vaccines and virotherapies.

Minor/typos

Line 255 live not life

Passage with the wrong wording has been removed

Line 509 cocktail not ocktail

Corrected

General addition without specific request:

In our initial manuscript submission we mentioned the generation and characterization of a dual prot insertion construct, however, no data were shown. We revised this part in the Results section and present the corresponding data in a new **Figure S6**.

Results Page 6 ; lines 136-145

*However, as a proof-of-concept to further minimize the risk for potential reversion to wild-type VSV, a tandem insertion was made combining the PR dimer insertions into P and L. We had confirmed feasibility of this approach by generating a recombinant VSV with two fluorescent protein insertions, one in P and one in L²⁴. Based on this observation, a recombinant VSV with PR dimer insertions in P and L (VSV-P-Lprot-GFP) was generated (**Figure S6a,b**). This virus also showed PI dependency, expressing GFP and generating small plaques only in the presence of APV (**Figure S6d**). However, the dual switch construct was significantly attenuated (> 2 log reduction in virus titers) compared to single insertion variants and was not further investigated in this study.*

Reviewers' Comments:

Reviewer #1:

Remarks to the Author:

The authors have addressed all issues raised by the reviewers to my satisfaction. Hence I can now recommend acceptance of this conceptually really interesting work for Nat Commun.

Reviewer #2:

Remarks to the Author:

my questions and request for changes have been adequately addressed